# UGC: UNIVERSAL GRAPH COARSENING

## ABSTRACT

In the era of big data, graphs have emerged as a natural representation for intricate relationships. However, graph sizes often become unwieldy, leading to storage, computation, and analysis challenges. A crucial demand arises for methods that can effectively downsize large graphs while retaining vital insights. Graph coarsening seeks to simplify large graphs while maintaining essential features. Most published methods are suitable for homophilic datasets, limiting their universal use. We propose Universal Graph Coarsening (UGC), a framework equally suitable for homophilic and heterophilic datasets. UGC integrates node attributes and adjacency information, leveraging the dataset's heterophily factor and is a first ever linear time-complexity framework. Results on benchmark datasets demonstrate that UGC preserves spectral similarity while coarsening. In comparison to state of the art methods, UGC is 4x to 15x faster, has lower eigen-error, and yields superior performance on downstream processing tasks even at 70% coarsening ratios.

## 1 INTRODUCTION

Graphs have emerged as highly expressive tools to represent diverse structures and knowledge in various fields such as social networks, bioinformatics, transportation, and natural language processing. They are essential for tasks like community detection, drug discovery, route optimization, and text analysis. With the growing importance of graph-based solutions, dealing with large graphs has become a challenge. Graph coarsening, a widely used technique to simplify graphs while retaining vital information, making them more manageable for analysis Kumar et al. (2020). It has been applied successfully in various tasks Hendrickson & Leland (1995); Karypis & Kumar (1999); Kushnir et al. (2006); Dhillon et al. (2007); Wang et al. (2014). Preserving the structural information of the graph is crucial in graph coarsening algorithms to ensure the fidelity of the coarsened graphs. A high-quality coarsened graph retains essential features and relationships, enabling accurate results for downstream tasks. Additionally, computational efficiency is equally vital for scalability, as large-scale graphs are common in real-world applications. Efficient coarsening method should ensure that the reduction in graph size does not come at the expense of excessive computation time but existing graph coarsening methods often face trade-offs between scalability and the quality of the coarsened graph. Our method draws inspiration from hashing techniques, which provides us with advantages in terms of computational efficiency. As a result, our approach exhibits a linear time complexity, making it highly efficient even for large graphs.

Graph datasets often exhibit a blend of homophilic and heterophilic traits Zhu et al. (2020); Pei et al. (2020). GC has been widely explored on homophilic datasets, but, to the best of our knowledge, has never been applied on heterophilic graphs. We propose Universal Graph Coarsening $UGC$, an approach that works well on both. Figure 1 illustrates how UGC uses a graph's adjacency matrix as well as the node feature matrix. UGC relies on hashing, lending computational efficiency. UGC exhibits linear time complexity, enabling fast processing of large datasets. UGC enhances the performance of Graph Convolutional Networks (GCN) in classification tasks, indicating its suitability for downstream processing. UGC coarsened graphs retain essential spectral properties, and show low eigen error, hyperbolic error, and $\epsilon$-similarity measure. In a nutshell, UGC is fast, universally applicable, and information preserving.

**Outline:** The paper is organized as follows: Section 2 briefly describes the related background and our problem formulation. We explain our framework in Section 3. We then provide assurances for quality similarities for coarsened graph in Section 4. Section 5 demonstrates the effectiveness of our algorithm with extensive experimentation. And then finally, the conclusion in Section 6.

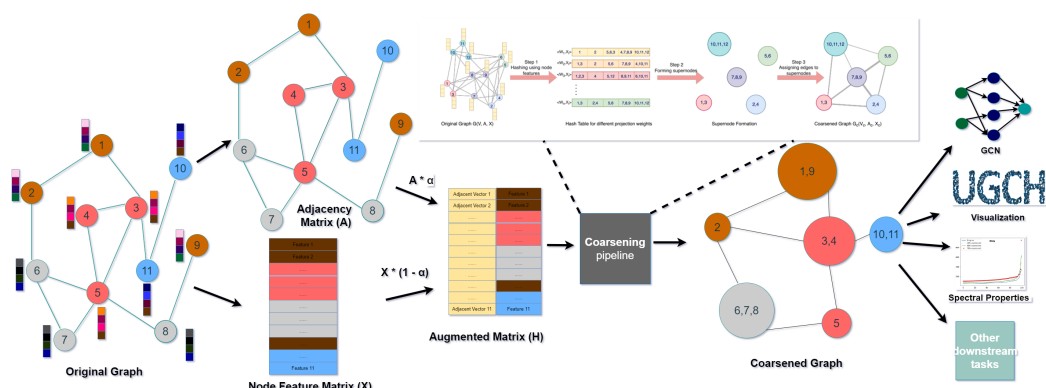

Figure 1: This figure illustrates our framework UGC, which depicts a) Generation of an augmented matrix by incorporating feature and adjacency matrices, while using heterophily measure $\alpha$. b) Coarsening the original graph using augmented features via Hashing. c) Validating reduced graph's quality and its utilization in subsequent downstream tasks.

## 2  BACKGROUND AND PROBLEM FORMULATION

A graph $G = (V, E, A, X)$ where $V = \{v_1, v_2, \cdots, v_{|N|}\}$ is the set of $N$ vertices, $E$ is the set of edges $(v_i, v_j) \subseteq (V \times V)$. $A \in \mathbb{R}^{N \times N}$ is the adjacency matrix. A non-zero entry $A_{ij}$ indicates there is an edge between nodes $i$ and $j$. The $i$-th row of the feature matrix $X \in \mathbb{R}^{N \times d}$, vector $X_i \in \mathbb{R}^d$, is the vector of features associated with the $i$-th node of $G$. The degree matrix $D$ is diagonal, with $D = diag(D_{i,i})$, where $D_{ii} = \sum_j A_{ij}$. The Graph Laplacian matrix $L = D - A$ Kipf & Welling (2016). $L \in \mathbb{R}^{N \times N}$ is a Laplacian matrix if it belongs to the set $S_L = \big\{ L \in \mathbb{R}^{N \times N} | L_{ji} = L_{ij} \leq 0, \ \forall i \neq j; L_{ii} = -\sum_{j \neq i} L_{ij} \big\}$.

**2.1  Problem Formulation**  The objective is to reduce an input Graph $G$ into a new graph $G_c(\tilde{V}, \tilde{E}, \tilde{A}, \tilde{X})$, with $n$-nodes and $\tilde{X} \in \mathbb{R}^{n \times d}$ where $n << N$ nodes. The GC problem may be thought of as learning a coarsening matrix $C \in R^{N \times n}$, which is a linear mapping from $V \to \tilde{V}$. A linear mapping ensures that similar

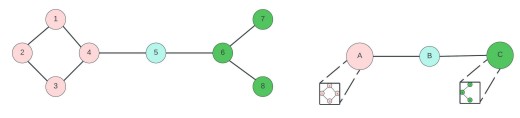

Figure 2: Toy graph coarsening example.

nodes in $G$ are mapped to the same super-node in $G_c$, s.t. $\tilde{X} = C^T X$. Every non-zero entry in $C_{ij}$ in $C$ denotes the merger of the $i^{th}$ node of $G$ to the $j^{th}$ supernode. This $C$ matrix belongs to the following set:

$$S = \left\{ C \in \mathbb{R}^{N \times n}, C_{ij} \in \{0, 1\}, \|C_i^T\| = 1, \langle C_i, C_j \rangle = 0, \forall i \neq j, \langle C_l, C_l \rangle = d_i, \|C_i\|_0 \geq 1 \right\} \quad (1)$$

where $d_i$ is the degree of the $i^{th}$-node. The condition $\langle C_i, C_j \rangle = 0$ ennsures that each node of $G$ is mapped to a unique super-node. The constraint $\|C_i\|_0 \geq 1$ requires that each super-node contains at least one node. Consider the 8-node graph in Fig. 2. Nodes 1, 2, 3, and 4 are mapped to supernode **A**, while nodes 6, 7, and 8 are mapped to supernode **C**. Hence, the loading matrix $C$ is given as Equation 2.

$$C^T = \begin{bmatrix} 1 & 1 & 1 & 1 & 0 & 0 & 0 & 0 \\ 0 & 0 & 0 & 0 & 1 & 0 & 0 & 0 \\ 0 & 0 & 0 & 0 & 0 & 1 & 1 & 1 \end{bmatrix} \quad (2)$$

**2.2  Homophilic and Heterophilic datasets**  Graph datasets may demonstrate homophily and heterophily properties Zhu et al. (2021); Du et al. (2022); McPherson et al. (2001); Shalizi & Thomas (2011); a heterophily factor $0 \leq \alpha \leq 1$ may be used to denote the degree of heterophily. $\alpha$ is calculated as the fraction of edges between nodes of different classes. Homophily refers to the tendency of nodes to be connected to other nodes of the same class or type, while heterophily signifies the tendency of nodes to connect with nodes of different classes. A strongly heterophilic graph ($\alpha \to 1$) has most edges between nodes of different classes, suggesting a diverse network with

mixed interactions. Conversely, weak heterophily, or strong homophily ($\alpha \to 0$) occurs in networks where nodes predominantly connect with others of the same class.

**2.3 Locality Sensitive Hashing** Locality Sensitive Hashing (LSH) is a linear time, efficient similarity search technique for high dimensional data Indyk & Motwani (1998); Kulis & Grauman (2009b); Buhler (2001); Satuluri & Parthasarathy (2012). It maps high dimensional vectors to lower dimensions, while ensuring that similar vectors collide with high probability. LSH uses a family of hash functions to map vectors to buckets, enabling fast retrieval and similarity search. It has found applications in image retrieval Kulis & Grauman (2009a), data mining Ravichandran et al. (2005), and similarity search algorithms Chum et al. (2007). LSH is defined as

**Definition 2.1** *Let $d$ be a distance measure, and let $d_1 < d_2$ be two distances. A family of functions $\mathcal{F}$ is said to be $(d_1, d_2, p_1, p_2)-$sensitive if for every $f \in \mathcal{F}$ the following two conditions hold*

- *If $d(x, y) \leq d_1$ then probability $[f(x) = f(y)] \geq p_1$*

- *If $d(x, y) \geq d_2$ then probability $[f(x) = f(y)] \leq p_2$*

UGC uses LSH with a set of random projectors to map similar nodes to the same super node. The projection is computed as $h_i(x) = \left\lfloor \frac{<x, a_i> - b_i}{w} \right\rfloor$, where $a_i$ is randomly selected from a distribution (such as Gaussian); $x$ represents the original high dimensional data sample, and $w$ is the width of each quantization bin.

**2.4 Related Works** The literature is replete with graph reduction methods, that aim to shrink a graph through deleting nodes by using vertex selection, re-combination schemes, or aggregation. Loukas proposed advanced spectral graph coarsening algorithms based on local variation, to preserve the original graph's spectral properties Loukas (2019). Two variants, *viz.* edge-based local variation and neighborhood-based local variation, select contraction sets with small local variation in each stage, but have limitations in achieving arbitrary coarsening levels Loukas (2019). Heavy edge matching (HEM), determines the contraction family by computing a maximum-weight matching based on the weight of each contraction set Dhillon et al. (2007); Ron et al. (2010). The Algebraic Distance method calculates the weight of each candidate set using an algebraic distance measure Ron et al. (2010); Chen & Safro (2011). The affinity method, inspired by algebraic distance, uses a vertex proximity heuristic Livne & Brandt (2011). The Kron reduction method Dorfler & Bullo (2013) was originally proposed for electrical networks, but is too slow for for large networks. Most of the above mentioned methods ignore node features during graph coarsening, and are usually memory intensive. UGC efficiently utilizes node features to achieve fast coarsening.

## 3 PROPOSED FRAMEWORK UNIVERSAL GRAPH COARSENING (UGC)

The proposed UGC framework comprises three main components, i) First, obtaining an augmented feature matrix containing both node feature and the structural information, ii) locality-sensitive hashing to derive the coarsening matrix, iii) finally, obtaining the coarsened graph adjacency matrix and coarsened features.

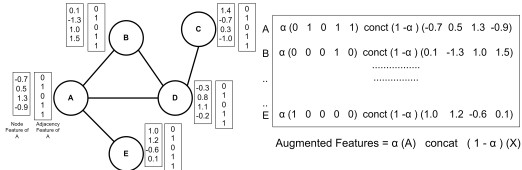

Figure 3: A toy example the computation of augmented features of a given graph.

**3.1 Augmented Feature** The concepts of homophily and heterophily in graphs are primarily concerned with whether the edges between two nodes align with their respective features. Therefore, in order to create a universal framework for graph coarsening suitable for all, it is important to consider both the node-level features and the structure-level features, such as the neighborhood structure of any given node, together. In this regard we create an augmented feature matrix $F$, where each node's feature vector $X_i$ is augmented with its one-hot adjacency vector $A_i$, with the heterophily hyperparameter $\alpha$ balancing the emphasis between feature and adjacency information. The augmented feature vector of each node is given by $F_i = \left\{ (1 - \alpha) \cdot X_i \oplus \alpha \cdot A_i \right\}$.. Figure 3 provides a toy example of the augmented feature vector. While larger graphs may result in long vectors, efficient implementations, and sparse tensor

methods may alleviate this hurdle. A motivating example demonstrating the need for augmented features while doing graph coarsening is discussed in the Appendix A.8, See figure 11.

**3.2 Proposed framework UGC** Let $F_i \in \mathbb{R}^d$ represent node $v_i$'s augmented feature vector. Let $P \in \mathbb{R}^{L \times d}$ and $b \in \mathbb{R}^L$ be the hashing matrices used in UGC, with $L$ denoting the number of projectors. The hash value given to the graph node is its most frequent hash index generated across all hash functions. Node $v_i$'s hash value is $h_i = maxOccured\{\lfloor \frac{1}{r} * (P \cdot X_i + b) \rfloor\}$, where $r$ is the bin-width. This hyperparameter controls the coarsened graph $G_c$'s size. Empirically, we find that increasing $r$ shrinks $G_c$. All nodes assigned the same hash value map to the same supernode in $G_c$.

---

**Algorithm 1** UGC: Universal Attributed Graph Coarsening via Hashing

---

**Require:** Input $G(V, A, X)$, $L \leftarrow$ # of Projectors, $r \leftarrow$ bin-width, $\alpha \leftarrow$ heterophilic factor, $n \leftarrow$ # of supernodes.
1: $F = (\alpha)(X) \oplus (1 - \alpha)(A)$ is now our augumented feature matrix.
2: **for** every projector $\ell \in \{1, 2, ...., L\}$ **do**
3:    $h^\ell \leftarrow N$ size array for Hash indices, $P^\ell \leftarrow d$ dimensional Projection weight, $b^\ell \leftarrow$ bias
4: **end for**
5: **for** $\ell \in \{1, 2, ...., L\}$ **do**
6:    $P_i^\ell \sim U[0, 1] \; \forall i \in \{1, 2, 3, ....., d\}, b^\ell \sim U[-r, r]$
7:    $h_i^\ell \leftarrow \lfloor \frac{1}{r} \times (\sum_{j=1}^{j=d}(F_i^j \times P_j^\ell) + b^\ell) \rfloor \; \forall i \in \{1, 2, 3, ....., N\}$
8: **end for**
9: $h_i \leftarrow maxOccurence\{h_i^\ell; \ell \in \{1, 2, 3, ..., L\}\} \forall i \in \{1, 2, 3, ....., N\}$
10: $\pi \leftarrow$ Dictionary mapping every node in $G$ to supernode $\in \{1, 2, ..., n\}$ in $G_c$
11: **for** every node v in V **do**
12:    $C[v, \pi[v]] \leftarrow 1$
13: **end for**
14: $A_c(i, j) \leftarrow \sum_{(u \in \pi^{-1}(\tilde{v}_i), v \in \pi^{-1}(\tilde{v}_j))} A_{uv}, \forall i, j \in \{1, 2, ..., n\}$
15: $F_c(i) \leftarrow \frac{1}{|\pi^{-1}(\tilde{v}_i)|} \sum_{u \in \pi^{-1}(\tilde{v}_i)} F_u, \forall i \in \{1, 2, ..., n\}$
16: return $G_c(V_c, A_c, F_c)$, $n$

---

When $G$ is coarsened to $G_c$, any supernode pair $(\tilde{v}_i, \tilde{v}_j)$, where $\tilde{v}_i, \tilde{v}_j \in G_c$, are joined by an edge, if there exists at least one node $u \in G$ that maps to $\tilde{v}_i$, that shares an edge with at least one node $v \in G$ that is mapped to $\tilde{v}_j \in G_c$. The weight of each edge in $G_c$, denoted by $A_c(ij)$, embodies the cumulative edge strength between the corresponding supernodes. This weight is determined by the sum of $A_{uv}$ for all $(u, v)$ pairs where $u \in \tilde{v}_i$ and $v \in \tilde{v}_j$. The features of supernodes are derived as the average of constituent node features: $F_c(i) = \frac{1}{|(\tilde{v}_i)|} \sum_{u \in (\tilde{v}_i)} F_u$. Notably, the label of a supernode is assigned based on the most prevalent class within it. The reader may like to refer to Algorithm 1 for the steps in UGC. A matrix $C \in \{0, 1\}^{N \times n}$ represents the partition of nodes, with $N$ as the count of nodes in the initial graph. The element $C_{ij}$ equals 1 if vertex $v_i$ is associated with supernode $\tilde{v}_j$. Crucially, every node is assigned a unique $h_i$ value, ensuring an exclusive mapping to a supernode. This constraint harmonizes with the formulation of supernode guarantees at least one node per supernode. Thus, each row of $C$ contains only one non-zero entry, leading to orthogonal columns. This matrix $C$ satisfies the conditions specified in equation 1.

**3.3 Edge Assignment and bin-width** By utilizing $C$, we calculate the adjacency matrix $A_c$ for $G_c$ by $A_c = AC$. As each super-edge consolidates multiple edges from the original graph, the total edge count in the coarsened graph is notably fewer than $m$. This results in a considerably sparser adjacency matrix $A_c$ compared to $A$. The parameter bin-width $r$ decides the size of the coarsened graph $G_c$. For a particular coarsening ratio $R$, we find the corresponding $r$ by divide and conquer approach on the real axis, which is similar to binary search. Algorithm 2 shows the method by which we find the $r$ for any given $R$ for $G_c$. Figure 8 shows the relation of $r$ with $R$ for two datasets: Cora & Coauthor CS. It is observed that the $R$ increases as the $r$ increases. For each dataset Bin-width finder is a hyper-parameter that needs to be run only once, and hence it is not included in the reported time complexity.

**3.4 Time Complexity Analysis of UGC** UGC achieves an efficient $O(NLd)$ time complexity, where N is the node count, L is the number of projectors, and d is the feature vector dimension. Key contributions come from three phases: hashing nodes(Line 1-7), aggregating bins(Line 8-12),

and computing supernode features(Line 14- 18). This analysis demonstrates UGC's scalability for large-scale graph coarsening tasks while maintaining vital structural characteristics. For detailed anaylsis please refer Appendix A.10.

## 4 QUALITY OF THE COARSENED GRAPH

It is desired that coarsened graph $G_c$ is similar to the original graph $G$. In this section, we have introduced various measures to assess the quality of coarsened graphs from multiple perspectives. Even though each of these characteristics has a unique sense of similarity, coarsening is better when these error levels are lower across all of them.

**Spectral similarities:** REE compares the Laplacian matrices of the coarsened and the original graph, as it measures the similarity between the eigenspace of $G$ and $G_c$. A low value of REE is desired for higher spectral similarity. REE is defined as $REE(L, L_c, k) = \frac{1}{k} \sum_{i=1}^{k} \frac{|\tilde{\lambda}_i - \lambda_i|}{\lambda_i}$ where $\lambda_i$ and $\tilde{\lambda}_i$ are top $k$ eigenvalues of Laplacian original ($L$) and Laplacian coarsened graph($L_c$) matrix respectively. Please refer to Section 5.3 for results.

**Structural similarities:** When moving from a $G_c$ representation back to the $G$, we quantify the disparity between the original and projected data using the Hyperbolic Error (HE). HE indicates the structural similarity between $G$ and $G_c$ with the help of a lifted matrix along with the feature matrix $X$ of the original graph. HE Bravo Hermsdorff & Gunderson (2019) defined as $HE = arccosh(\frac{||(L - L_{\text{lift}})X||_F^2 ||X||_F^2}{2 trace(X^T LX) trace(X^T L_{\text{lift}} X)} + 1)$ where $L$ and $X \in \mathbb{R}^{N \times d}$ are the Laplacian, and $X$ is the feature matrix of the original input graph, and $L_{\text{lift}}$ is the lifted Laplacian matrix defined in Loukas & Vandergheynst (2018) as $L_{\text{lift}} = CL_cC^T$ where $C \in \mathbb{R}^{n \times N}$ is the coarsening/loading matrix and $L_c$ is the Laplacian of $G_c$.

**LSH ensuring intra supernode similarity:** The LSH family used in our framework is based on p-stable distributions, which ensures that the probability of two nodes going to the same supernode is directly related to the distance between their features (augmented features for UGC). This is expressed in the below Theorem from Indyk & Motwani (1998):

**Theorem 4.1** *The probability that two nodes $v$ and $u$ will collide and go to a supernode under a hash function drawn uniformly at random from a 2-stable distribution is inversely proportional to $c = ||v - u||_2$.*

$$p(c) = Pr_{a,b}[h_{a,b}(v) = h_{a,b}(u)] = \int_0^r \frac{1}{c} f_p\left(\frac{t}{c}\right)\left(1 - \frac{t}{r}\right) dt \tag{3}$$

The above result will be used to check if the mapping of the nodes to the supernode ensures the distance and hence similarity property.
*Proof*: The proof is deferred to appendix A.4

$\epsilon$-**similarity:** The previously discussed metrics primarily utilize either the adjacency matrix or the feature matrix independently. However, as we are using augmented feature matrix for hashing, it becomes essential to introduce a measure that takes both feature and adjacency information into account simultaneously. The Dirichlet energy (DE), employed to measure the smoothness of graph signals, is defined utilizing the graph Laplacian matrix $L \in S_L$ and the feature matrix $X$ as $DE(L, X) = tr(X^T LX) = -\sum_{i,j} L_{ij}||x_i - x_j||^2$. Loukas (2019) suggested to use the following induced semi-norms $||X||_L = \sqrt{x^T Lx}$, $||X_c||_{L_c} = \sqrt{x_c^T L_c x_c}$ and defined $\epsilon$-similarity via Theorem 4.2.

**Theorem 4.2** *Given a Graph $G$ and a coarsened graph $G_c$ they are said to be $\epsilon$ similar if there exists some $\epsilon \geq 0$ such that:*

$$(1 - \epsilon)||X||_L \leq ||X_c||_{L_c} \leq (1 + \epsilon)||X||_L \tag{4}$$

*where $L$ and $L_c$ are the Laplacian matrices of $G$ and $G_c$ respectively.*

*Proof:* The proof is deferred in the Appendix A.6

This definition is directly applied to our framework UGC as our loading matrix $C$ discussed in section 2.1 follows constraints discussed in equation 1 making $L_c$ an laplacian matrix($L_c = C^T L C$). The coarsened graph $G_c$ generated through UGC exhibits a high degree of similarity, within the range of $\epsilon$, to the original graph $G$. It has been empirically demonstrated that this coarsened representation performs exceptionally well across various downstream tasks. Nonetheless, for the purpose of achieving a more refined and stringent upper limit on the permissible $\epsilon$ value (where $\epsilon \leq 1$), a potential step involves introducing modifications to the feature learning procedure of the supernodes $G_c$.

**Bounded $\epsilon-$similarity:** It is important to note that the $\epsilon$-similarity measure introduced in Loukas (2019) does not incorporate features. Instead, it relies on the eigenvector of the Laplacian matrix to compute similarity, which limits its ability to capture the characteristics of the associated features along with the graph structure. Kumar et al. (2023) re-define Theorem 4.2 but with feature matrix. Once we get the loading matrix $C$ using UGC as discussed in Section 3.2 we used $F_c(i) = \frac{1}{|\pi^{-1}(\tilde{v_i})|} \sum_{u \in \pi^{-1}(\tilde{v_i})} F_u$ to learn the new feature-vectors of our super-nodes, where $F_u$ is the augmented feature vector of node $u$ from $G$. To give a bound on the $\epsilon$ similarity($\leq 1$) we suggest to update $F_c$ ($\tilde{F}$) by minimizing the term

$$\min_{\widetilde{F}} f(\widetilde{F}) = \mathrm{tr}(\widetilde{F}^T C^T L C \widetilde{F}) + \frac{\alpha}{2} \|C\widetilde{F} - F\|_F^2 \qquad (5)$$

*Proof:* The proof is deferred in the Appendix A.7.

Using the new update rule of $\|\tilde{F}\|_{L_c}$ we have $\tilde{F}_{L_c} \leq \|F\|_L$, we get $\epsilon = \frac{|\|F\|_L - \|\widetilde{F}\|_{L_c}|}{\|F\|_L} \leq 1$ where $\epsilon \leq 1$ refer Kumar et al. (2023) for more details.

**Scalable Training of GNNs:** In addition to above comprehensive analysis, we have rigorously assessed the practical utility of our coarsened graph in downstream tasks. The coarsened graph is employed for training a Graph Convolutional Network (GCN) model, which is subsequently utilized to make predictions on the test data of the original nodes. It is important to emphasize that the intricacies of the Graph Neural Network (GNN) architectures fall outside the scope of this paper, as our primary focus remains the validation of our coarsened graphs. To this end, we adopted a standardized 2-layer GCN model, with detailed parameters conforming to the guidelines outlined in Huang et al. (2021). This study advocated the utilization of coarsened graphs for scalable GNN training. Results are discussed in Section 5.5.

## 5 EXPERIMENTS

In this section, we conduct extensive experiments to evaluate the proposed UGC against the existing graph coarsening algorithms. The conducted experiments showcase the performance of our framework concerning computational efficiency, preservation of spectral properties, and potential extensions of the coarsened graph into real-world applications.

**5.1 Experimental Protocol** We compare our proposed algorithm with the following coarsening algorithms as discussed in Section 2.4, two variation methods based on edges and neighborhood Loukas & Vandergheynst (2018), Algebraic Distance Chen & Safro (2011), Affinity Livne & Brandt (2011), Heavy Edge Dhillon et al. (2007); Ron et al. (2010)

Table 1: Summary of the datasets. H.R shows heterophily factor.

| Data | Nodes | Edges | Features | Class | H.R |
|------|-------|-------|----------|-------|-----|
| Cora | 2,708 | 5,429 | 1,433 | 7 | 0.81 |
| Citeseer | 3,327 | 9,104 | 3,703 | 6 | 0.74 |
| DBLP | 17,716 | 52,867 | 1,639 | 4 | 0.82 |
| CS | 18,333 | 163,788 | 6,805 | 15 | 0.80 |
| PubMed | 19,717 | 44,338 | 500 | 3 | 0.80 |
| Phy. | 34,493 | 247,962 | 8,415 | 5 | 0.93 |
| Flickr | 89,250 | 899,756 | 500 | 7 | 0.31 |
| Reddit | 232,965 | 114.615M | 602 | 41 | 0.75 |
| Yelp | 716,847 | 13.954M | 300 | 100 | |
| Texas | 183 | 309 | 1703 | 5 | 0.09 |
| Cornell | 183 | 295 | 1703 | 5 | 0.3 |
| Film | 7600 | 33544 | 931 | 5 | 0.22 |
| Squirrel | 5201 | 217073 | 2089 | 5 | 0.22 |
| Chameleon | 2277 | 36101 | 2325 | 5 | 0.25 |

and Kron Dorfler & Bullo (2013). We show that time complexity wise UGC is substantially better than all of these methods. We have also shown that the quality of the $G_c$ is on par with these algorithms. Our experiments cover widely adopted benchmarks, including Cora, CiteSeer, Coauthor CS, Coauthor Physics, DBLP, and PubMed. Additionally, UGC effectively coarsens large datasets like Flickr, Reddit, and Yelp, previously challenging for existing techniques. We also present datasets like squirral, Chameleon, Texas, Film, and Wisconsin, characterized by dominant heterophilic factors. Table 1 provides comprehensive dataset details, including the heterophilic factor (H.R). All the

Table 2: Summary of run-time in seconds averaged over 5 runs to reduce the graph to 50% coarsening ratios. It can be seen that for massive datasets where all methods are not even able to run, UGC is giving a coarsened graph in a matter of seconds.

| Data/Method | Cite. | PubMed | DBLP | Physics | Reddit | Yelp | Squirrel | Cham. | Cor. | Texax | Film |
|---|---|---|---|---|---|---|---|---|---|---|---|
| Var. Neigh. | 8.72 | 24.38 | 22.79 | 58.0 | OOM | OOM | 33.26 | 12.2 | 1.34 | 0.63 | 27.67 |
| Var. Edges | 7.37 | 18.69 | 20.59 | 67.16 | OOM | OOM | 46.45 | 12.65 | 1.31 | 0.76 | 26.6 |
| Var. Cliq. | 9.8 | 61.85 | 38.31 | 69.80 | OOM | OOM | 28.91 | 10.55 | 1.56 | 1.14 | 33.04 |
| Heavy Edge | 1.41 | 12.03 | 8.39 | 39.77 | OOM | OOM | 18.08 | 5.41 | 1.62 | 1.17 | 11.79 |
| Alg. Dist | 1.55 | 10.48 | 9.67 | 46.42 | OOM | OOM | 18.03 | 5.24 | 1.58 | 0.81 | 12.65 |
| Affinity GS | 2.53 | 168.3 | 110.9 | 924.7 | OOM | OOM | 20.00 | 5.83 | 1.81 | 1.24 | 20.65 |
| Kron | 1.37 | 0.63 | 7.09 | 34.53 | OOM | OOM | 20.62 | 7.25 | 1.73 | 0.97 | 12.29 |
| UGC | **0.71** | **1.62** | **1.86** | **6.4** | **16.17** | **170.91** | **2.14** | **0.49** | **0.04** | **0.03** | **1.38** |

Table 3: Relative Eigen Error at 50% coarsening ratio

| Data/Method | Cite. | PubMed | DBLP | Physics | Reddit | Yelp | Squirrel | Cham. | Cor. | Texax | Film |
|---|---|---|---|---|---|---|---|---|---|---|---|
| Var. Neigh. | 0.180 | 0.108 | 0.117 | 0.273 | OOM | OOM | 0.871 | 0.657 | 0.501 | 0.391 | 32.87 |
| Var. Edges | 0.136 | 0.965 | 0.135 | 0.042 | OOM | OOM | 0.298 | 0.597 | 0.485 | 0.489 | 21.8 |
| Var. Cli. | 0.064 | 1.208 | 0.082 | 0.039 | OOM | OOM | 0.369 | 0.456 | 0.550 | 0.463 | 22.95 |
| Hea. Edge | 0.043 | 0.834 | 0.086 | 0.031 | OOM | OOM | 0.256 | 0.333 | 0.554 | 0.464 | 5.69 |
| Alg. Dist. | 0.111 | 0.403 | 0.047 | 0.117 | OOM | OOM | 0.245 | 0.413 | 0.552 | 0.465 | 5.71 |
| Aff. GS | 0.057 | 0.063 | 0.073 | 0.052 | OOM | OOM | **0.226** | 0.413 | 0.569 | 0.489 | 5.56 |
| Kron | **0.028** | 0.378 | 0.060 | 0.064 | OOM | OOM | 0.246 | 0.413 | 0.554 | 0.491 | 6.12 |
| UGC(fea.) | 0.340 | 0.179 | 0.145 | **0.016** | EOOM | EOOM | 13.8 | 7.594 | 0.420 | 0.534 | 9.83 |
| UGC(fea+Ad) | 0.070 | **0.004** | **0.004** | 0.018 | EOOM | EOOM | 0.546 | **0.429** | **0.215** | **0.204** | **0.075** |

experiments conducted for this work were performed on an Intel Xeon W-295 CPU and 64GB of RAM desktop using the Python environment.

**5.2 Experiments for Run-time analysis.** The primary focus of our algorithm's contribution lies in computational efficiency. The findings of this section provide empirical support for our assertions regarding time complexity. The time required to derive the coarsening matrix $C$ using UGC is summarized in Table 2. By referring to this Table, it becomes evident that UGC exhibits a remarkable advantage, surpassing all current algorithms across diverse datasets. Our model outperforms existing methods by a substantial margin. While other methods struggles at large datasets like physics, UGC is able to coarsen down massive datasets like Yelp, which was previously not possible. It should be emphasized that the time taken by UGC on the Reddit dataset which has $7\times$ more number of nodes compared to Physics is one-third the time taken by the fastest state-of-the-art methods on the Physics dataset. This Table also encompasses heterophilic datasets, computational efficacy is clearly visible with these datasets as well.

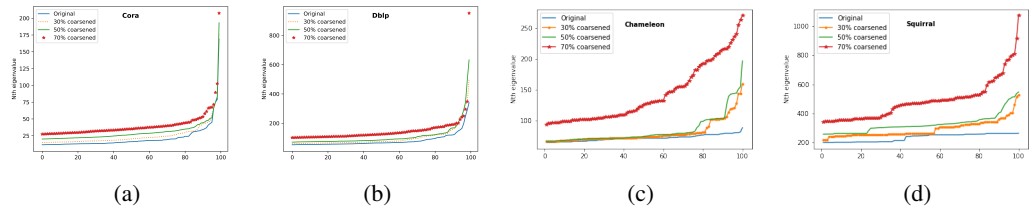

Figure 4: Top 100 eigenvalues of the original and coarsened graph at three different coarsening ratios: 30%, 50%, and 70%. We can observe that the spectral property is maintained across all coarsening ratios for all coarsened graphs. For a lower coarsening ratio, this approximation (REE) is more accurate.

**5.3 Spectral properties preservation.** As mentioned in section 4, we use relative eigen error (REE) and hyperbolic error (HE) as the evaluation metrics to measure the spectral and structural similarity of our coarsened graph. Eigenvalues preservation can be clearly seen in Figure 4 where we have plotted the top 100 eigenvalues of original and of coarsened graphs. We can see that spectral property is preserved even for 70% coarsened graph for most of the datasets. This approximation is

Table 4: Accuracy of GCN model when trained with 50% coarsen graph

| Data/Method | Cite. | Cora | CS | DBLP | PubMed | Physics | Squirrel | Cham. | Cor. | Texax | Film |
|---|---|---|---|---|---|---|---|---|---|---|---|
| Var.Neigh. | 69.54 | 79.75 | 87.90 | 77.05 | 77.87 | 93.74 | 19.67 | 20.03 | 52.49 | 34.51 | 15.67 |
| Var.Edges | 70.60 | 81.57 | 88.74 | **79.93** | 78.34 | 93.86 | 20.22 | 29.95 | 55.32 | 30.59 | 21.8 |
| Var.Clique | 68.81 | 80.92 | 85.66 | 79.15 | 73.32 | 92.94 | 19.54 | 31.92 | 58.8 | 33.92 | 20.35 |
| Heavy Edge | 71.11 | 79.90 | 69.54 | 77.46 | 74.66 | 93.03 | 20.36 | 33.3 | 54.67 | 29.18 | 19.16 |
| Alg. Dis. | 70.09 | 79.83 | 83.74 | 74.51 | 74.59 | 93.94 | 19.96 | 28.81 | **59.91** | 18.61 | 19.23 |
| Aff. GS | 70.70 | 80.20 | 87.15 | 78.15 | 80.53 | 93.06 | 20.00 | 27.58 | 54.06 | 21.18 | 20.34 |
| Kron | 69.00 | 80.71 | 85.35 | 77.79 | 74.89 | 92.26 | 18.03 | 29.1 | 55.02 | 31.14 | 17.41 |
| UGC(fea.) | 66.97 | 83.92 | 77.19 | 75.50 | **85.65** | 94.70 | 20.71 | 29.9 | 55.6 | 52.4 | 22.6 |
| UGC(fea+Ad) | **74.55** | **89.30** | **93.02** | 75.50 | 84.77 | **96.12** | **31.62** | **48.7** | 54.7 | **57.1** | **25.4** |

more accurate for a lower coarsening ratio i.e smaller the graph the bigger is Relative Eigen Error (REE). The Relative Eigen Error for all approaches across all datasets is shown in Table 3 for a fixed 50% coarsening ratio. REE value of UGC is comparable with most of the algorithms. Although we also have coarsened Graphs for large datasets like Yelp and Reddit, eigen error calculation for these datasets was out of memory so we have used EOOM while other methods fail to find even the coarsened graph hence the term OOM. Figure 5 illustrates how, for different techniques, the eigen error, hyperbolic error, and GCN accuracy change as the coarsening ratio is altered.

## 5.4 LSH similarity and $\epsilon$-bounded results
In our experiments, we empirically validated the equation 3. We achieved this validation by utilizing a distance matrix derived from node features. We examined if the distance between any node pair was below a specific threshold, and then using the partition matrix given by UGC we verified if they shared the same supernode or not. Our evaluation involved counting successful matches, where

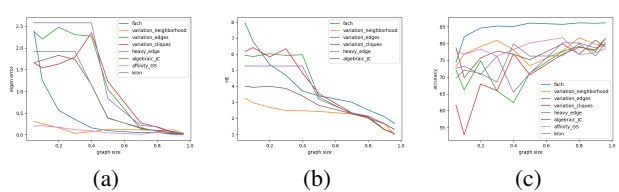

Figure 5: This figure shows the comparison of all graph coarsening methods in terms of REE, HE, and GCN accuracy on the PubMed dataset.

nodes belonged to the same supernode, and failures, where they did not. We subsequently calculated a probability measure based on these counts. Part a) and b) of Figure 6 plots this probabilistic function for two datasets, namely Cora and Citeseer as a function of distance between two nodes. Re-visting the Definition 2.1 for Cora dataset, we denote our LSH family as $\mathcal{H}(1,3,100,0.20)$. Part c) of Figure 6 which plots different values of $\epsilon$ at different coarsening ratios. As mentioned we got $\epsilon \leq 1$ similarity guarantees for the coarsened graph. Hence proving the bounded version Theorem 4.2.

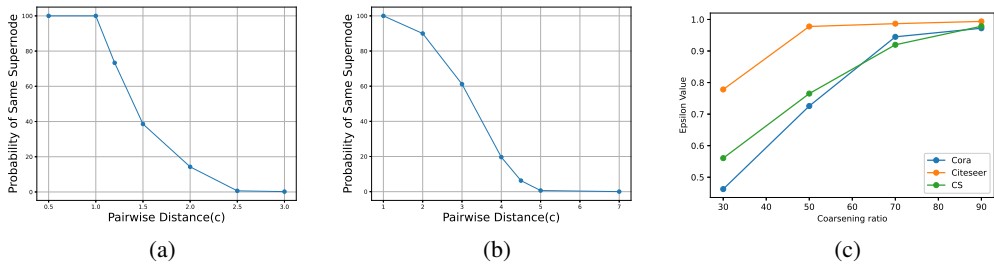

Figure 6: Figure a) Cora and b) Citeseer demonstrates the inverse relationship between the probability of two nodes belonging to the same supernode as distance between them increases. Figure c) plots the $\epsilon$ values( $\leq 1$) for Cora, Citeseer and CS datasets.

## 5.5 Scalable Training of Graph Neural Networks.
Graph neural networks (GNNs), tailored for non-Euclidean data Bruna et al. (2013); Chen et al. (2020); Defferrard et al. (2016), have shown promise in various applications Li & Goldwasser (2019); Paliwal et al. (2019). However, scalability

remains a challenge. Building on Huang et al. (2021), we investigate how our graph coarsening approach can enhance GNN scalability for training, bridging the gap between GNNs and efficient processing of large-scale data. UGC(feat) represents a specific scenario within our framework, wherein only the feature values are considered for hashing, thereby obtaining the mapping of supernodes. To comprehend the significance of incorporating the adjacency vector, we have added the results for both UGC(feat) and UGC(augmented feat).

**Experimental Details.** We employed a single hidden layer GCN model with standard hyperparameters values Kipf & Welling (2016). Coarsened data is used to train the GCN model and all the prediction is being done on original graph data. The learned weights on the coarsened graph $G_c$, are then used for making predictions on the original graph, $G$. The relation between coarsening ratio and accuracy can be clearly seen from the Table 6 i.e if we reduce the graph more and more we starts to see a slight decrease in accuracy values. Hence there will always be a trade-off when it comes to coarsening ratio and quality of reduced graph. We have included a Figure 7 which shows how much we have gained in computational time and what is the change in the accuracy values when compared to the existing best model for different datasets. Table 4 compares the accuracy among all the approaches with all datasets when they are coarsened down by 50%. We have used t-SNE van der Maaten & Hinton (2008) algorithm for visualization of predicted node labels shown in Figure 9. It is evident that even with 70% coarsened data training GCN model is able to maintain its accuracy. Very few of the data points are miss-classified(mostly outliers) when we increase our coarsening ratio to reduce the original graph.

**5.6 Gained performance on Heterophilic Graphs** Building upon the observations made in Table 3 and Table 4. Our methods, UGC(feat) and UGC(aug. feat.), demonstrate significant enhancements in node classification accuracy and REE values across heterophilic datasets. Comparing these results, we observe that traditional approaches exhibits moderate performance in terms of accuracy. However, when employing our method UGC(features), we achieve notable improvements in accuracy, surpassing the performance of these traditional approaches. Moreover, the true potential of our approach is unleashed when incorporating both features and adjacency information i.e augmented features. This combined approach showcases remarkable accuracy gains, outperforming all other methods by a considerable margin. Our method not only yields competitive results in the homophilic graph scenario but also exhibits substantial improvements when applied to heterophilic datasets.

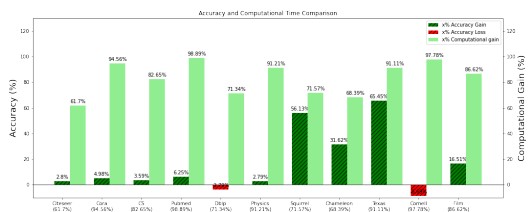

Figure 7: The figure illustrates the effectiveness of our algorithm concerning computational gains and accuracy improvement in comparison to the current state-of-the-art methods for each dataset. In the bar plot, dashed bars represent the gain or loss in accuracy when compared to the best-performing method, while plain bars indicate the computational gains achieved. All datasets are coarsened down by 50%

## 6 CONCLUSION

In this paper, we present a framework **UGC** for reducing a larger graph to a smaller graph. We use hashing of augmented node features inspired by Locality Sensitive Hashing (LSH). As expected, the benefits of LSH are also reflected in the proposed coarsening algorithm. UGC has linear time complexity with respect to the number of nodes. To the best of our knowledge, it is the fastest algorithm for graph coarsening. Through extensive experiments, we have also shown that our algorithm is not only fast but also preserves the spectrum and smoothness properties of the original graph. Furthermore, it is worth noting that UGC represents a first work in the domain of graph coarsening for heterophilic datasets. This framework addresses the unique challenges posed by heterophilic graphs and has demonstrated a significant increase in node classification accuracy following graph coarsening. This is a significant accomplishment, as it opens up new possibilities for training GNNs on large and complex graphs. In conclusion, we believe that our framework, is a major contribution to the field of graph coarsening and offers a fast and effective solution for simplifying large networks. Our future research goals include the exploration of different hash functions and novel applications for the framework.

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

# A  APPENDIX

**A.1  Stable Distribution**  A distribution $\mathcal{D}$ over $\mathcal{R}$ is called p-stable, if there exists p $\geq 0$ such that for any n real numbers $v_1....v_n$ and i.i.d. variables $X_1....X_n$ with distribution $\mathcal{D}$, the random variable $\sum_i v_i X_i$ has the same distribution as the variable $(\sum_i |v_i|^p)^{1/p} X$ where $X$ is a random variable with distribution $\mathcal{D}$ Datar et al. (2004). It is known Zolatarev (1983) that stable distributions exists for p $\in$ (0,2].

- *Cauchy distribution* $\mathcal{D}_c$, defined by the density function $c(x) = \frac{1}{\pi}\frac{1}{1+x^2}$, is 1-stable.

- *Gaussian (normal) distribution* $\mathcal{D}_g$, defined by the density function $g(x) = \frac{1}{\sqrt{2\pi}}e^{\frac{-x^2}{2}}$ is 2-stable.

However, it is known Chambers et al. (1976) that one can create p-stable random variables effectively from two independent variables distributed uniformly across [0,1] despite the lack of closed form density and distribution functions.

Stable distributions have diverse applications across various fields (see survey Nolan (2005) for details). In computer science, they are utilized for "sketching" high-dimensional vectors, as demonstrated by Indyk (Indyk (2006)). The key property of p-stable distributions, mentioned in the definition, enables a sketching technique for high-dimensional vectors. This technique involves generating a random vector **w** of dimension **d**, with each entry independently chosen from a p-stable distribution. Given a vector **v** of dimension d, the dot product $w \cdot v$ is also random variable. A small collection of such dot products, corresponding to different w's, is termed as the sketch of the vector v and can be used to estimate $||v||_p$ Indyk (2006). However, instead of using the sketch to estimate the vector norm we are using it to assign hash values to each vector. These values map each vector to a point on the real line, which is then splitted into equal width segments to represent buckets. If two vectors v and u are close, they will have a small difference between $l_p$ norms $\|v - u\|_p$, and they should collide with a high probability.

**A.2  Additional experiments for LSH scheme**  We have further validated our theoretical results through a secondary experiment. This LSH family which we discussed above says as the distance between two nodes increases, the likelihood of them being assigned to the same bin decreases, hence we will have more number of supernodes now. By increasing the bin-width, we can effectively reduce the number of supernodes. This phenomenon is evident when considering the average distance between node pairs in various graphs and the corresponding bin-width required to achieve a 30% coarsening ratio. The table below illustrates these findings:

The results in the table clearly demonstrate that as the average distance between nodes increases, the required bin-width also increases when maintaining the same coarsening ratio. This observation highlights the importance of considering the distance metric and bin-width selection during the graph coarsening process to effectively control the number of supernodes and achieve desired coarsening ratios. Figure 8 shows trend of coarsening ratio when we change bin-width.

Table 5: Average Distance and Bin-Width for 30% Coarsening

| Dataset | Average Distance | Bin-Width |
|---------|------------------|-----------|
| Citeseer | 7.748 | 0.0029 |
| Cora | 5.810 | 0.0021 |
| Dblp | 3.168 | 0.000068 |
| Pubmed | 0.540 | 0.000025 |

**A.3  Bin-width**  This section discusses the impact of bin-width on the coarsening ratio see Figure 8. Algorithm 2 outlines the procedure for determining the appropriate bin-width value that corresponds to a desired coarsening ratio.

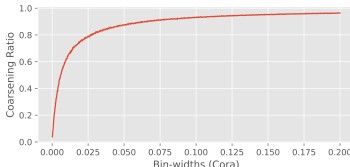 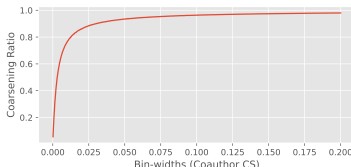

Figure 8: This figure shows the trend of coarsening ratio as the bin-width increases on two datasets: Cora and Coauthor CS.

---

**Algorithm 2** Bin-width Finder

---

**Require:** Input $G(V, A, X)$, $L \leftarrow$ # of Projectors, $R \leftarrow$ Desired Coarsening Ratio, $p \leftarrow$ precision of coarsening, $N \leftarrow$ # of nodes in the graph $G$, $n \leftarrow$ # of nodes in the graph $G_c$
**Ensure:** bin-width $h$
1: $r \leftarrow 1$, $ratio \leftarrow 1$
2: **while** $|c - ratio| > p$ **do**
3:    **if** $ratio > R$ **then**
4:       $r \leftarrow r * 0.5$
5:    **else**
6:       $r \leftarrow r * 1.5$
7:    **end if**
8:    _, $n \leftarrow$ UGC$(G, L, h, N)$
9:    $ratio \leftarrow (1 - \frac{n}{N})$
10: **end while**
11: return $r$

---

**A.4 Proof of Theorem 4.1** Let $f_p(t)$ denote the probability density function of absolute value of our stable distribution(Normal distribution), and let $c = ||v - u||_p$ for two node vectors v, u and r is the bin-width. Since we have a random vector w from our stable distribution, $v.w - u.w$ is distributed as cX where X is a random variable from our stable distribution. Therefore our probability function is

$$p(c) = Pr_{a,b}[h_{a,b}(v) = h_{a,b}(u)] = \int_0^r \frac{1}{c} f_p\left(\frac{t}{c}\right)\left(1 - \frac{t}{r}\right) dt \tag{6}$$

For a fixed bin-width $r$ the probability of collision decreases monotonically with $c = ||v - u||_2$. For Defination, 2.1 the hash family will be $(r_1, r_2, p_1, p_2)$-sensitive where $p_1 = p(1)$ and $p_2 = p(c)$ for $\frac{r_2}{r_1} = c$.

For 2-stable distribution $f_p(x) = \frac{2}{\sqrt{2\pi}} e^{-x^2/2}$. Equation 3 will be

$$p(c) = \frac{2}{\sqrt{2\pi}} \int_0^r \frac{1}{c} e^{-\left(\frac{1}{c}\right)^2/2} dt - \frac{2}{\sqrt{2\pi}} \int_0^r \frac{1}{c} e^{-\left(\frac{1}{c}\right)^2/2} \frac{t}{r} dt \tag{7}$$

$$= S_1(c) - S_2(c)$$

Note that $S_1(c) \leq 1$.

$$S_2(c) = \frac{2}{\sqrt{2\pi}} \cdot \frac{c}{r} \int_0^r e^{-\left(\frac{t}{c}\right)^2/2} \frac{t}{c^2} dt \tag{8}$$

$$S_2(c) = \frac{2}{\sqrt{2\pi}} \cdot \frac{c}{r} \int_0^{\frac{r^2}{(2c^2)}} e^{-y} dy \tag{9}$$

$$S_2(c) = \frac{2}{\sqrt{2\pi}} \cdot \frac{c}{r} [1 - e^{-\frac{r^2}{(2c^2)}}] \tag{10}$$

We have $p(1) = S_1(1) - S_2(1) \geq 1 - e^{\frac{r^2}{2}} - \frac{2}{\sqrt{2\pi}r} \geq 1 - \frac{A}{r}$, for some constant A > 0. This implies that the probability that u collides with v is at least $(1 - \frac{A}{r}) \approx e^{-A}$. Thus the algorithm is correct

with the constant probability.
If $c^2 \leq \frac{r^2}{2}$, then we have

$$p(c) \leq 1 - \frac{2}{\sqrt{2\pi}} \frac{c}{r} (1 - \frac{1}{e}) \tag{11}$$

**A.5  Application of coarsened graph for GNNs**  This section contains additional results related to the scalable GNN training. Figure 10 shows the GNN training pipeline. Figure 9 shows Visualization of GCN predicted nodes when training is done using the coarsened graph. Table 6 illustrates a direct correlation between accuracy and coarsening ratio. As the graph is coarsened more aggressively, we observe a corresponding decrease in the classification accuracy of the GCN model.

Table 6: We report the accuracy of GCN on node classification after coarsening by UGC at different ratios.

| Ratio/Data | Cite. | Cora | CS | DBLP | Pub. | Phy. |
|---|---|---|---|---|---|---|
| 30 | 74.89 | 89.30 | 93.02 | 75.50 | 85.65 | 96.70 |
| 50 | 74.55 | 89.30 | 93.02 | 75.50 | 84.77 | 96.12 |
| 70 | 71.27 | 84.63 | 88.29 | 74.82 | 80.57 | 92.43 |

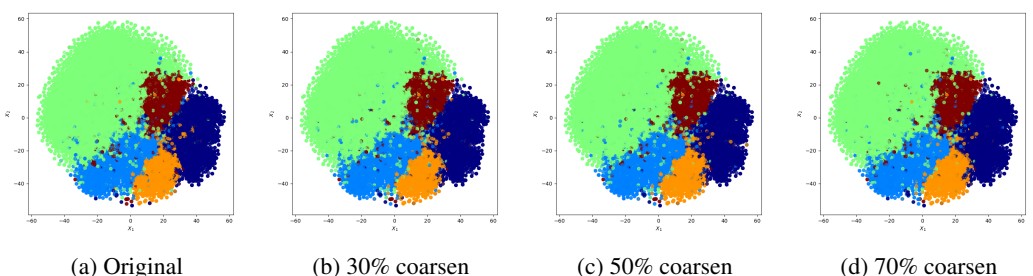

(a) Original      (b) 30% coarsen      (c) 50% coarsen      (d) 70% coarsen

Figure 9: Visualization of GCN predicted nodes when training is done using the coarsened graph.

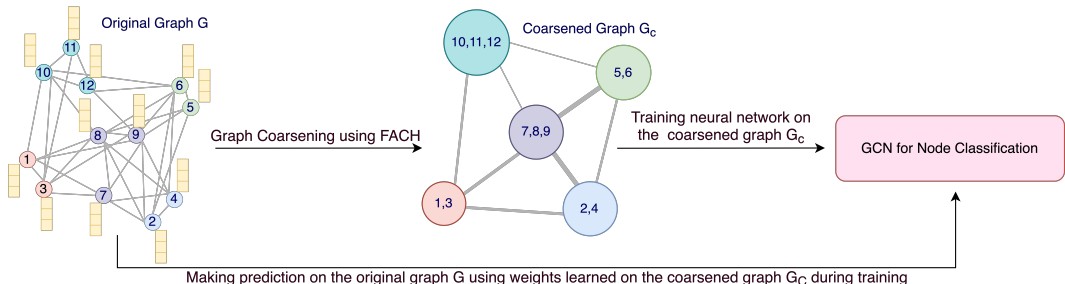

Figure 10: GCN training pipeline

**A.6  Proof of $\epsilon-$similarity Theorem 4.2**

**Theorem A.1** *Given a Graph $G$ and a coarsened graph $G_c$ they are said to be $\epsilon$ similar if there exists some $\epsilon \geq 0$ such that:*

$$(1 - \epsilon)\|x\|_L \leq \|x\|_{L_c} \leq (1 + \epsilon)\|x\|_L \tag{12}$$

*where $L$ and $L_c$ are the Laplacian matrices of $G$ and $G_c$ respectively.*

**Proof:** Let S be defined such that $L = S^T S$. By triangle inequality:

$$\|\|x\|_L - \|x_c\|_{L_c} = \|\|Sx\|_2 - \|SP^+Px\|_2 \tag{13}$$

$$\leq \|Sx - SP^+Px\|_2 = \|x - \tilde{x}\|_L \leq \|x\|_L \tag{14}$$

## A.7 Proof of Bounded Theorem 4.2

$$\min_{\widetilde{F}} f(\widetilde{F}) = \text{tr}(\widetilde{F}^T C^T L C \widetilde{F}) + \frac{\alpha}{2}\|\widetilde{F}C - F\|_F^2 \tag{15}$$

Here we have replaced partition matrix $P$ with $C = P^\dagger$ as discussed in Section 2.1. The above equation is a convex optimization problem from which we get a closed form solution by putting the gradient w.r.t to $\tilde{F}$ equal to zero.

$$2C^T L C \widetilde{F} + \alpha C^T (C\widetilde{F} - F) = 0, \tag{16}$$

Update rule for $\widetilde{F}$

$$\widetilde{F}^{t+1} = (\frac{2}{\alpha}C^T L C + C^T C)^{-1} C^T F \tag{17}$$

Using F, $\widetilde{F}$ and re-writing Theorem 4.2 as

$$\|\|F\|_L - \|\widetilde{F}\|_{L_c}\| = |\sqrt{tr(F^T L F)} - \sqrt{tr(\widetilde{F}^T L_c \widetilde{F})}| \tag{18}$$

As $L$ is a positive semi-definite matrix we can write $L = S^T S$ using Cholesky's decomposition and by writing $L_c = C^T L C$ we get,

$$= |\sqrt{tr(F^T S^T S F)} - \sqrt{tr(\widetilde{F}^T C^T S^T S C \widetilde{F})}| \tag{19}$$

$$= \|\|SF\|_F - \|SP^\dagger PF\|\|_F \tag{20}$$

$$\leq \|\|SF - SP^\dagger PF\|\|_F \tag{21}$$

$$\leq \epsilon\|F\|_L \tag{22}$$

Using the new update rule of $\|\tilde{F}\|_{L_c}$ we have $\tilde{F}_{L_c} \leq \|F\|_L$, we get

$$\epsilon = \frac{\|\|F\|_L - \|\widetilde{F}\|_{L_c}\|}{\|F\|_L} \leq 1 \tag{23}$$

where $\epsilon \leq 1$ refer Kumar et al. (2023) for more details. See Figure 6 which plots different values of $\epsilon$ at different coarsening ratios. As mentioned for fixed values of $\alpha$ we got $\epsilon \leq 1$ similarity gaurntees for the coarsened graph.

**A.8 Importance of Augmented Features** See Figure 11 which showcase the importance of considering the augmented feature vector. It can be seen from the figure that when coarsened using Augmented features supernodes have more intra-node similarity.

**A.9 Additional Results** Table 7 and Table 8 contains Run-time and REE results for Cora, CS and Flickr datasets.

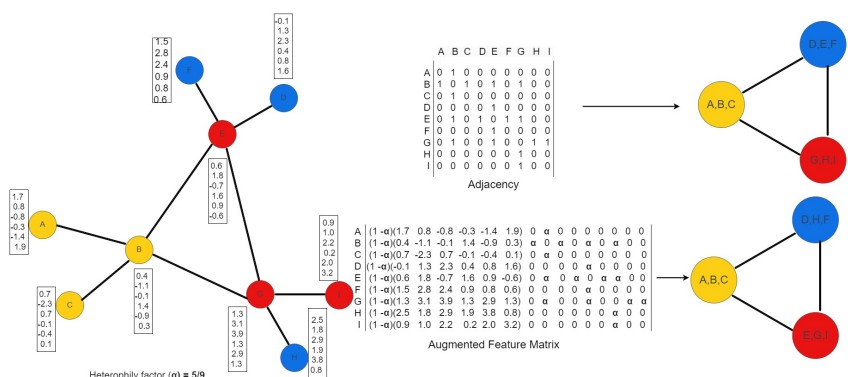

Figure 11: This figure highlights the significance of the augmented vector and showcases coarsening outcomes, specifically when coarsening is performed solely using the adjacency or feature matrix compared to when the augmented matrix is taken into account.

Table 7: Additional results for Run Time.

| Ratio/Data | Cora | CS | Flickr |
|---|---|---|---|
| Var. Neigh. | 6.64 | 23.43 | OOM |
| Var. Edges | 5.34 | 16.72 | OOM |
| Var. Cliques | 7.29 | 24.59 | OOM |
| Heavy Edge | 0.70 | 7.50 | OOM |
| Alg. Distance | 0.93 | 9.63 | OOM |
| Affinity GS | 2.36 | 169.05 | OOM |
| Kron | 0.63 | 5.81 | OOM |
| UGC | **0.41** | **3.1** | **8.9** |

**A.10 Detailed Time Complexity Analysis** We have three phase for our framework. In the first phase(Algo 1 Line 1-7), we can see Line 7 is driving the complexity of the algorithm where we multiply two $F \in \mathbb{R}^{N \times d}$ and $W \in \mathbb{R}^{L \times d}$ matrices which results to $O(NLd)$. In the second pass, the supernodes for the coarsened graphs are constructed with the help of the accumulation of nodes in the bins. The main contribution of UGC is up to these two phases i.e., Line 1-10. Till now, time-complexity is $O(NLd) \equiv O(NC)$ where $C$ is a constant. Hence, the time complexity for getting the partition matrix is $O(N)$.

In the third phase, Line 14-15, we calculate the adjacency and features of the supernodes of the coarsened graph $G_c$. For this we iterate over the edges of the original graph and use the edge nodes along with the surjective mapping $\pi : V \to V_c$ to increment the weight of the corresponding edge between the supernodes in $G_c$. The computational cost of this operation is $O(m)$, where $m$ is the number of edges in the original graph, and this is a one time step. Indeed, the overall time complexity of all three phases combined is O(N+m) where m is the number of edges. However, it's important to note that the primary contribution of UGC lies in the process of finding the partition matrix whose

Table 8: Additional results for Eigen Error

| Ratio/Data | Cora | CS | Flickr |
|---|---|---|---|
| Var. Neigh. | 0.1211 | 0.2488 | OOM |
| Var. Edges | 0.1293 | 0.0498 | OOM |
| Var. Cliques | 0.0850 | **0.0263** | OOM |
| Heavy Edge | 0.0713 | 0.0467 | OOM |
| Alg. Distance | 0.1079 | 0.0872 | OOM |
| Affinity GS | 0.0950 | 0.0633 | OOM |
| Kron | **0.0695** | 0.0564 | OOM |
| UGC | 0.1309 | 0.0570 | 0.0153 |

Table 9: UGC is model-agnostic

| Model/Data | Cora | Pubmed | Physics | Squirrel |
|:---:|:---:|:---:|:---:|:---:|
| GCN | 89.30 | 84.77 | 96.12 | 31.62 |
| GraphSage | 69.39 | 85.72 | 94.49 | 61.23 |
| GIN | 67.23 | 84.12 | 85.15 | 44.72 |
| GAT | 74.21 | 84.37 | 92.60 | 48.75 |

time complexity is O(N). We have compared the partition matrix computational of all other methods with ours.

**A.11 UGC is model-agnostic** While our initial validation utilized GCN to assess the quality of our coarsened graph but our framework is not bound to any specific graph convolutional network (GCN) architecture. We extended our evaluations to include other prominent graph neural network models. Results from three diverse models, namely GraphSage, GIN (Graph Isomorphism Network), and GAT (Graph Attention Network), have been incorporated into our analysis. We have provided empirical evidences in Table 9. These additional experiments demonstrate the robustness and model-agnostic nature of our framework. We believe this flexibility further enhances the applicability and utility of our proposed framework in various graph-based applications.

