# OpenReview forum: "UGC: UNIVERSAL GRAPH COARSENING"
_ICLR.cc/2024/Conference — Submitted to ICLR 2024_

### Official Review · Reviewer_Nt9N · 2023-10-28

**Soundness:** 3 good
**Presentation:** 3 good
**Contribution:** 3 good
**Rating:** 6
**Confidence:** 4

**Summary:**

This paper gives a graph coarsening algorithm based on random projections. The nodes are repeatedly hashed using a random projections, and then assigned to vertices in the smaller graphs via a majority scheme.

Guarantees of this reduction scheme are given via properties of random projections. The effectiveness of this coarsening scheme are then experimentally measured, demonstrating good efficiency, preservation of spectral properties, and in training of graph neural networks.

**Strengths:**

The random projection based scheme is natural, and is well known to be among the most efficient possible. The experiments are quite extensive, and demonstrate a lot of useful and intriguing properties about this coarsening scheme.

**Weaknesses:**

Some of the formal derivations were a bit difficult to parse, e.g. in equation (1) on page 2, what is the <C_l, C_l> term? I also mistook the \forall i \neq j before it to be for this term too because it included a d_i.

Also, in theorem 4.2, it's not clear what the role of x is.

After equation (5), it's not clear what the `proof in Appendix A.7' is for (is it missing a theorem statement here?)

**Questions:**

I had difficulties finding a concise summary of the theoretical guarantees (in terms of the graph Laplacian) proven about this coarsening scheme. Would it be possible to point to a 'main theorem' that's proven?

---

> ### Author Response · Authors · 2023-11-17
> **Eagerly waiting for your feedback**
>
> Thank you for reviewing the paper and providing feedback! We have responded below to individual quotes from your review.
>
> **Ques 1)** Some of the formal derivations were a bit difficult to parse, e.g. in equation (1) on page 2, what is the <C_l, C_l> term? I also mistook the \forall i \neq j before it to be for this term too because it included a d_i.
>
> **Ans 1)** We thank the reviewer for bringing this to our attention. We have utilized the condition $C_l, C_l = d_l$ to denote the degree of super-nodes. We appreciate the reviewer's feedback and have addressed the concern by removing the comma before $\forall i \neq j$ to enhance clarity and avoid any potential confusion in the main manuscript.
>
> **Ques 2)** In theorem 4.2, it's not clear what the role of x is.
> After equation (5), it's not clear what the 'proof in Appendix A.7' is for (is it missing a theorem statement here?). I had difficulties finding a concise summary of the theoretical guarantees (in terms of the graph Laplacian) proven by this coarsening scheme. Would it be possible to point to a 'main theorem' that's proven?
>
> **Ans 2)**
>
> In theorem 4.2 $x \in R^d$ represents the feature vector of the node, while $X \in R^{N * d}$ represents a node features matrix of all the nodes.
>
> **Theorem 4.2 :($\epsilon-similarity$)**
> Given a Graph $G(L,X)$ and a coarsened graph $G_c(L_c,X_c)$, they are said to be $\epsilon$ similar if there exists some $\epsilon \geq 0$ such that:
>
>
>
> $(1-\epsilon)\lVert X\rVert_{L}\leq\lVert X_c\rVert_{L_c}  \leq (1 + \epsilon) \lVert X\rVert_L$
>
> where $L$ and $L_c$ are the Laplacian matrices of $G$ and $G_c$ respectively. The coarsened graph$G_c({L_c,X_c}$) and features learned as an outcome of UGC satisfy this $\epsilon$ similarity.
>
> In the section **Bounded $\epsilon$-similarity**, we have further improved this $\epsilon$ similarity result by relearning the coarsened features via imposing the smoothness condition mentioned in Equation 5. As a consequence of it we are able to obtain a better bound on $\epsilon$, i.e., $\epsilon\sim(0,1]$ which is summarized in the below Theorem
>
>
> **Updated Theorem:**
> Given a Graph **$G(L,X)$** we learn a coarsened Graph via UGC **$G_c(L_c, X_c)$**, and then we relearn enhanced features as, $\tilde{X}$ by enforcing the smoothness condition. The original graph $G(L,X)$ and coarsened graph $G_c(L_c,\tilde{X})$ are $\epsilon$ similar with $0 < \epsilon \leq 1$
>
> $(1-\epsilon)\lVert X\rVert_L\leq \lVert\tilde{X}\rVert_{L_c}\leq (1+\epsilon)\lVert X\rVert_L$
>
> The proof regarding this Theorem is deferred in Appendix A.7.
>
> If the reviewer deems it beneficial, we are open to putting this discussion in the main paper or Appendix.
>
> For theoretical guarantees(in terms of the graph Laplacian) we refer reviewers to this updated version of Theorem 4.2.

---

> > ### Comment · Reviewer_Nt9N · 2023-11-18
> >
> > Thank you for addressing these issues, and the additional information / pointers.

---

> > > ### Author Response · Authors · 2023-11-20
> > >
> > > Thank you for positively responding to the paper. **We would appreciate if you could reconsider your scores on the paper in the light of the responses to the comments of all reviewers, in case you feel it deserves to be in.**

---

### Official Review · Reviewer_rN8j · 2023-10-31

**Soundness:** 1 poor
**Presentation:** 1 poor
**Contribution:** 2 fair
**Rating:** 3
**Confidence:** 4

**Summary:**

The paper presents a strategy called UGC to coarsen an attributed graph to a smaller graph, while preserving certain desirable traits (e.g., certain spectral properties). The algorithm uses locality sensitive hashing to operate in a fast manner, and is empirically tested in a number of tasks.

**Strengths:**

The problem is certainly relevant and the idea of employing locality sensitive hashing appears novel.
I believe there are several interesting ideas encapsulated in this paper, which appear however to be not fully developed

**Weaknesses:**

* The paper is written in a somewhat haphazard way: the notation and introduced concepts remain often unclear and the paper lacks a clear structure and organization in my opinion.

* Problem formulation -- what precisely is the mathematical objective that UGC tries to achieve? This is not clearly stated.
The set S does not define the set of indicator matrices the authors seem to have in mind (there are matrices that fulfill those constraints that are not indicator matrices). Why introduce it this way? Also C is supposed to be N times n, yet the example given is n times N.

* The discussion on heterophily appears out of the blue -- it is not clear what relevance it has to the paper and the technique used.

* The discussion on related work is pretty mixed, but it is not always clear on what these relations are build. There is a whole literature on network summarization, which appears to be largely ignored that is much closer to the type of problems discussed in this paper.

* The notation and decsription in section 3.1 is unclear to me. What are the asterisks as opposed to \cdot denote? Figure 3 does not really help much either as it is rather cryptic.

* Section 3.2. and 3.3 are not well written either. There is an algorithm but the idea of it is hardly explained and the intuition remains completely absent in my opinion.

* Section 4 appears to be a long list of quality criteria, whose relative merits and selection is never discussed. Symbols appear that have not been introduced before etc.

* It is not really clear what questions the experimental session tries to answer and why. For instance, yes, run-time can be important, but only in conjunction with some other assessment of the quality of the coarsening -- what does 50% coarsening even mean for the other methods? How would you coarsen a graph to 50% by kron reduction -- there are many options. It is not clear what we are comparing here..

Overall the paper appears to have been put together in a rushed manner. There are many typos and grammatical mistakes throughout, the organization is not clear, and the key messages get lost in details.

**Questions:**

see weaknesses.

---

> ### Author Response · Authors · 2023-11-17
> **Eagerly waiting for your feedback**
>
> Thank you for reviewing the paper and providing feedback! We have responded below to individual quotes from your review.
>
> **Ques 1)** Problem formulation -- what precisely is the mathematical objective that UGC tries to achieve? This is not clearly stated. The set S does not define the set of indicator matrices the authors seem to have in mind (there are matrices that fulfill those constraints that are not indicator matrices). Why introduce it this way? Also C is supposed to be N times n, yet the example given is n times N.
>
> **Ans 1)**
> We have tried to elucidate the notation. To clarify, S is not a set of indicator matrices. It is the set of coarsening matrices. A coarsening matrix maps each node of the original graph to a node in the coarsened graph. Each node in the coarsened graph may be thought of as a supernode, that subsumes many nodes of the original graph. Hence, entries in a coarsening matrix are {1,0} indicating node numbers; each entry ($C_{ij}$) indicates the $i^{th}$ node number in the original graph paired with the $j^{th}$ node number in the coarsened graph.
>
> ***Problem formulation*** This manuscript presents a novel approach to coarsening all types of graphs, including those that largely follow a homophily assumption, and others that depart substantially. The coarsening task attempts to to reduce a given graph $G(V,E,A,X)$ with N nodes, into a new graph $G_c(\tilde{V},\tilde{E},\tilde{A}, \tilde{X})$, with $n$-nodes and $\tilde{X} \in \mathbb{R}^{n \times d}$ where $n<< N$ nodes. The coarsened graph  $G_c$ is summarized from the original graph $G$ by using a coarsening matrix, i.e. the Graph Coarsening problem is that of learning a coarsening matrix $C \in R^{N\times n}$. Every non-zero entry $C_{ij}$ in $C$ denotes the merger of the $i^{th}$ node of $G$ to the $j^{th}$ supernode. This $C$ matrix belongs to the following set S:
>
> $$  S =  \{ C \in \mathbb{R}^{N \times n}, \langle C_i,C_j \rangle = 0 \quad \forall i \neq j, \langle C_l,C_l \rangle = d_i, \|C_i\|_0 \geq 1  \} $$
> This equation becomes important to impose the next two discussed properties on the *“C”* matrix; the condition $\langle C_i, C_j \rangle = 0$ ensures that each node of $G$ is mapped to a unique super-node. The constraint $\|C_i\|_0 \geq 1$ requires that each super-node in $G_c$ contains at least one node of $G$.
>
> We thank reviewer for pointing out we have modified $C$ to $C^T$ in the example.
>
>
> **Ques 2)** The discussion on heterophily appears out of the blue -- it is not clear what relevance it has to the paper and the technique used.
>
> **Ans 2)** We clarify the point. One novelty of our work is its applicability to a wide range of datasets; this includes those satisfying homophily assumptions, as well as those that violate it, such as graphs characterized by heterophily. Since the literature on graph coarsening largely assumes homophily, this point is relevant and important in the context of our work. Hence, the concept of heterophily is mentioned in the abstract, introduction, and explicitly discussed in Section 2.2.
>
> **Ques 3)** The discussion on related work is pretty mixed, but it is not always clear on what these relations are build. There is a whole literature on network summarization, which appears to be largely ignored that is much closer to the type of problems discussed in this paper.
>
> **Ans 3)**  The manuscript cites recent literature on Graph Coarsening [1,2,3]. References were chosen from S-O-T-A methods to provide a baseline for UGC on two criteria. These include speed, and the ability to handle datasets that do not exhibit homophily. We might have missed some relevant references, it would be great if reviewers can point out those network summarization references relevant to this work. Certainly, more references could be included. However, with regard to speed, UGC is compared with the fastest methods published.
>
> [1] Loukas, Andreas. "Graph Reduction with Spectral and Cut Guarantees." J. Mach. Learn. Res. 20.116 (2019): 1-42.
>
> [2] Huang, Zengfeng, et al. "Scaling up graph neural networks via graph coarsening." Proceedings of the 27th ACM SIGKDD conference on knowledge discovery & data mining. 2021.
>
> [3] Kumar, Manoj, Anurag Sharma, and Sandeep Kumar. "A Unified Framework for Optimization-Based Graph Coarsening." Journal of Machine Learning Research 24.118 (2023): 1-50.
>
>
>
> **Ques 4)** The notation and description in section 3.1 is unclear to me. What are the asterisks as opposed to \cdot denote? Figure 3 does not really help much either as it is rather cryptic.
>
> **Ans 4)**  We thank the reviewer for pointing out the confusion, we have modified the equation in section 3.1 to $$ F_i = \{(1 - \alpha) \cdot  X_i\oplus \alpha \cdot A_i \}$$ where $\oplus$ represents the concatenation. And $\cdot$ is the multiplication. Figure 3 is used to visualize this augmented feature.

---

> > ### Author Response · Authors · 2023-11-17
> >
> > **Ques 5)** Section 4 appears to be a long list of quality criteria, whose relative merits and selection is never discussed. Symbols appear that have not been introduced before etc.
> >
> > **Ans 5)** In order to better highlight the efficacy of our approach, namely UGC, we have used multiple metrics to establish our claims. Section 4 discusses them in detail in order to explain the relative merits of each metric in the context of graph coarsening.
> >
> > ***Section 4 quality measures***
> > We concur with the reviewer that coarsening is only meaningful in the context of quality metrics being met. In consonance with that objective, in Section 5, we systematically evaluate the coarsened graph on multiple metrics.
> >  Eigen Error (REE) quantifies the dissimilarity between the eigen spaces of the original and coarsened graphs. HE measures the dissimilarity between the reconstructed and original graphs. We have used LSH similarity to present a probabilistic bound in Theorem 4.1. Finally, we have utilized $\epsilon$-similarity, that establishes a quality bound by considering both the features and adjacent vectors of the original graph. For the reader’s convenience, the metrics have been discussed in detail in Section 4.
> >
> > **Ques 6)** It is not really clear what questions the experimental session tries to answer and why. For instance, yes, run-time can be important, but only in conjunction with some other assessment of the quality of the coarsening -- what does 50% coarsening even mean for the other methods? How would you coarsen a graph to 50% by kron reduction -- there are many options. It is not clear what we are comparing here.
> >
> > **Ans 6)**
> >
> > In the experimental section, we have three main objectives
> >
> > **a)** To generate a coarsened graph that closely resembles the original graph
> >
> > **b)** Check the quality of our coarsened graph
> >
> > **c)** Coarsened graph application on downstream tasks.
> >
> > Coarsened graph is achieved by our framework **UGC**, Section 5.3 and Section 5.4 rigorously assess the quality of the coarsened graph using the comprehensive set of quality measures discussed in Section 4. Additionally, Section 5.5 demonstrates the effectiveness of graph coarsening in scaling Graph Neural Network (GNN) training using the coarsened graph.
> >
> > Notably, training a GNN model with the coarsened graph has yielded significant improvements in node classification accuracy, particularly on heterophilic data. This is discussed in detail in Section 5.6.
> >
> > Mentioned methods employ a 50% coarsening strategy, where, for example, if we initially have 1000 nodes in graph $G$, the application of each specific method results in a reduction to 500 nodes in the coarsened graph
> > $G_c$.
> >
> > Andreas Loukas’s widely cited work “Graph Reduction with Spectral and Cut Guarantees”,[1], states on pages 23, that “Kron reduction is an effective way to half the graph size but can result in poor approximation otherwise. If one is willing to sacrifice in terms of efficiency, Kron reduction effectively reduces the graph size by a factor of two (with the exception of the yeast graph).”. Tables 1 and 2 on page 24, provide results for reduction by factors of upto 70%, using various methods including Kron reduction. We cite this to point out that 50% coarsening using Kron reduction is not unusual. We reproduce Table 1 from Loukas’s paper here for convenience.
> >
> > | Coarsening Ratio\Method |heavy edge | local var.     | kron   |
> > | --------   | -------- | -------- | -------- |
> > | 30%        | 0.284    | 0.123    | 0.054         |
> > | 50%  | 1.069    | 0.460    |  1.321    |
> > | 70%        | 5.126    | 3.920    | 1.865    |
> >
> > REE reported in [1] for yeast dataset. Please see page 24 Table 1 and Table 2 in [1] for more details.
> >
> > [1] Loukas, Andreas. "Graph Reduction with Spectral and Cut Guarantees." J. Mach. Learn. Res. 20.116 (2019): 1-42.

---

> > > ### Author Response · Authors · 2023-11-20
> > >
> > > Dear Reviewer,
> > >
> > > Thanks again for taking the time to review our paper and for your feedback!
> > > May we enquire if all the concerns you raised have been adequately addressed? We greatly appreciate your prompt response, as the rebuttal deadline is approaching.

---

> ### Comment · Reviewer_rN8j · 2023-11-20
> **reponse**
>
> re A1) This is exactly what I would have referred to as indicator matrices. To say it once more in different terms: It appears the authors have a set of 0/1 matrices in mind; however, the characterization of the set is such that it contains many matrices that are not 0/1 -- for instance we may simply multiply the matrix C in the example by -1 and it would still be in the set...
> I believe the correct definition of the matrices C could be given much more succinctly in other terms.
>
> The current problem formulation the authors point out just states what the approach does (coarsen a graph) but it does not say what it aims to do in quantitative terms; e.g., is the spectrum supposed to be (approximately) conserved? some cut metric? the eigenvectors? There is still no mathematical problem statement here, in my opinion.
>
> re A2) homophily is not a property of a graph, but of a generative process that creates a graph (it is a causal model for the formation of links and it cannot be inferred from observational data alone -- see [1,2]) it is thus still not clear to me what the authors really mean. My guess is that they want to say that prior methods for graph coarsening do not show good performance according to some measure (but which one?) if the graph considered is created according to a process that is not homophilic.
>
> [1] McPherson, Miller, Lynn Smith-Lovin, and James M. Cook. "Birds of a feather: Homophily in social networks." Annual review of sociology 27.1 (2001): 415-444.
> [2] Shalizi, Cosma Rohilla, and Andrew C. Thomas. "Homophily and contagion are generically confounded in observational social network studies." Sociological methods & research 40.2 (2011): 211-239.
>
> There are also no citations given in this paragraph, so it is unclear what methods the author refer to when they say, those do not work for non-homophilic graphs.
>
> re A3) the mentioned papers all have a clear problem formulation and objective, which the current paper is still lacking in my opinion. Again graph coarsening is an ill-defined term and as long as the authors do not clearly specify what properties the coarsened graph is supposed to preserve precisely
> There are even surveys on graph summarization available already like Liu, Yike, et al. "Graph summarization methods and applications: A survey." ACM computing surveys (CSUR) 51.3 (2018): 1-34
>
> re A4) Thanks for the clarification
>
> re A5) What is not fully clear here is why the authors believe that their method would perform well in terms of these metrics (in terms of the algorithm design). That these are possibly quality criteria is quite clear, but how do they relate to the proposed approach?
>
> re A6) What does "resemble" and "quality" mean precisely here -- this is the question. In mathematical, quantifyable terms.
> Loukas work has a clear objective, already stated in the title of his work!
>
>
> Let me iterate my point here from my initial review: I think the problem area and potentially the algorithm the authors propose is interesting; however, the exposition and the discussion of the idea, motivation and algorithmic guarantees appears too vague to me to recommend acceptance here.

---

> > ### Author Response · Authors · 2023-11-21
> >
> > **Ans re A1)**
> >
> > Thank you for pointing out the confusion, and we appreciate the opportunity to clarify the definition of the matrices C in our paper.
> >
> > Note that the elements of C encodes node super-node mapping, where $i$ is the the node index of node mapped to the $j$th super node, and thus  $$C_{ij} \geq 0 $$ More specifically,  $C\in {0,1}^{N\times n}$ but with specific structure such that $C$ defines a valid partition matrix (mapping matrix).
> >
> > These mapping properties are
> > i) each node should be mapped to a super node uniquely  means the cardinality of the column of $$ ||C^T_i|| = 1\; and <C_i,C_j> = 0 $$ ii) no-supernode should be empty which means $$ ||C_i||_0 \geq 1 $$
> >
> > For example, matrices 2, 3 and 4 all belong to this set $$C_{ij}\geq 0 $$ but are not a valid partition matrix.
> >
> > A strict set of matrix satisfying the mapping property is written in form of Equation 1,
> >
> > $$  S =  \{ C \in \mathbb{R}^{N \times n},  C_{ij} \in \{0,1\}, \|C^T_i \| = 1, \langle C_i,C_j \rangle = 0 \quad \forall i \neq j, \langle C_l,C_l \rangle = d_i, \|C_i\|_0 \geq 1  \} $$
> >
> > here $C^T_i$ means column vectors of $C^T$.
> >
> > To illustrate, consider the example below:
> >
> > Matrix 1
> >
> > | 1 0 0 |
> >
> > | 1 0 0 |
> >
> > | 1 0 0 |
> >
> > | 0 1 0 |
> >
> > | 0 0 1 |
> >
> > Matrix 2  &nbsp; &nbsp;&nbsp;&nbsp;&nbsp;   Matrix 3 &nbsp; &nbsp;&nbsp;&nbsp;&nbsp;   Matrix 4
> >
> > | 1 0 1 |  &nbsp; &nbsp;&nbsp;&nbsp;&nbsp; | 0 1 0 | &nbsp; &nbsp;&nbsp;&nbsp;&nbsp;  | 0 0 0 |
> >
> > | 1 0 0 |  &nbsp; &nbsp;&nbsp;&nbsp;&nbsp; | 0 1 0 |  &nbsp; &nbsp;&nbsp;&nbsp;&nbsp; | 1 0 0 |
> >
> > | 1 0 0 | &nbsp; &nbsp;&nbsp;&nbsp;&nbsp;  | 0 1 0 |  &nbsp; &nbsp;&nbsp;&nbsp;&nbsp; | 1 0 0 |
> >
> > | 1 0 0 |  &nbsp; &nbsp;&nbsp;&nbsp;&nbsp; | 0 0 1 | &nbsp; &nbsp;&nbsp;&nbsp;&nbsp;  | 0 1 0 |
> >
> > | 0 1 0 | &nbsp; &nbsp;&nbsp;&nbsp;&nbsp;  | 0 0 1 |  &nbsp; &nbsp;&nbsp;&nbsp;&nbsp; | 0 0 1 |
> >
> > Equation 1 will make sure only the Matrix 1 is a valid partition matrix while Matrix 2,3,4 are not partition matrices.
> >
> >
> >
> > Furthermore, our definition of the partition matrix is not arbitrary but aligns with established literature. We request the reviewer to refer to Equation 1 on page 4 of [1], where a similar characterization of the partition matrix is employed.
> >
> > [1] Kumar, Manoj, Anurag Sharma, and Sandeep Kumar. "A Unified Framework for Optimization-Based Graph Coarsening." Journal of Machine Learning Research 24.118 (2023): 1-50.
> >
> >
> > **Ans re A2)**
> > We appreciate the reviewer's insightful comment and would like to provide further clarification on the concept of homophily in our paper.
> >
> > While homophily is indeed a property of the generative process that creates a graph, we want to emphasize that the term "homophily" in our paper refers to the tendency of nodes with similar characteristics to be connected. This observation is grounded in a body of literature [2,3,4,5] where the datasets and this definition used in our paper are standard and extensively discussed.
> >
> > Our intention  is to highlight that, to the best of our knowledge, there is limited existing work specifically addressing attributed graph coarsening, as evidenced by the literature [1]. Building upon this foundation, our method introduces modifications to the feature matrix by incorporating information from the adjacent vector. Notably, our approach represents the first-ever attempt to address coarsening in the context of these heterophilic datasets. In Table 1 we have given H.R indicating the heterophily factor.
> >
> > Thank you for pointing out the missing citations we have added them to main paper.
> >
> > [1] Kumar, Manoj, Anurag Sharma, and Sandeep Kumar. "A Unified Framework for Optimization-Based Graph Coarsening." Journal of Machine Learning Research 24.118 (2023): 1-50.
> >
> > [2] Chien, Eli, et al. "Adaptive universal generalized pagerank graph neural network." arXiv preprint arXiv:2006.07988 (2020).
> >
> >
> > [3] Cavallo, Andrea, et al. "GCNH: A Simple Method For Representation Learning On Heterophilous Graphs." arXiv preprint arXiv:2304.10896 (2023).
> >
> > [4] Zhu, Jiong, et al. "Beyond homophily in graph neural networks: Current limitations and effective designs." Advances in neural information processing systems 33 (2020): 7793-7804.
> >
> > [5] Du, Lun, et al. "Gbk-gnn: Gated bi-kernel graph neural networks for modeling both homophily and heterophily." Proceedings of the ACM Web Conference 2022. 2022.

---

> > > ### Author Response · Authors · 2023-11-21
> > >
> > > **Ans re A3)**
> > > We thank the reviewer for pointing out the confusion. We have tried to refine the problem formulation.
> > >
> > > **Problem formulation:**
> > >
> > > Given a graph $G(X,L)$ where $X \in R^{N \times d}$ is a feature matrix and $L \in R^{N \times N}$ is a Laplacian matrix of graph $G$, we aim to learn a partition matrix $C \in R^{N \times n}$ and  coarsened Laplacian matrix $L_c$ which satisfies . The matrix C defines the coarsened Laplacian matrix as $$L_c = C^T L C$$ $L_c$ will be used to get a coarsened graph $G_c(X_c,L_c)$ where $X_c \in R^{n \times d}$ and $L_c \in R^{n \times n}$ represent the feature and Laplacian matrices of $G_c$. Our objective is to obtain a $G_c$ approximates $G$ such that it satisfies certain properties mentioned in Section 4.
> > >
> > > These properties are:
> > > i) $G_c$ is said to be $\epsilon-$similar to $G$
> > >
> > > **Theorem 4.2 :($\epsilon-similarity$)**
> > > Given a Graph $G(L,X)$ and a coarsened graph $G_c(L_c,X_c)$, they are said to be $\epsilon$ similar if there exists some $\epsilon \geq 0$ such that:
> > >
> > >
> > >
> > > $(1-\epsilon)\lVert X\rVert_{L}\leq\lVert X_c\rVert_{L_c}  \leq (1 + \epsilon) \lVert X\rVert_L$
> > >
> > > where $L$ and $L_c$ are the Laplacian matrices of $G$ and $G_c$ respectively. The coarsened graph$G_c({L_c,X_c}$) and features learned as an outcome of UGC satisfy this $\epsilon$ similarity.
> > >
> > >
> > > ii) LSH ensuring intra supernode similarity: It ensures that the probability of two nodes going to the same supernode is directly related to the distance between their features
> > >
> > > **Theorem 4.1 :($LSH similarity$)**
> > > The probability that two nodes $v$ and $u$ will collide and go to a supernode under a hash function drawn uniformly at random from a 2-stable distribution is inversely proportional to $c = || v - u ||_2$.
> > >
> > > Given by equation
> > >
> > > $$p(c)=P r_{a, b}\left[h_{a, b}(v)=h_{a, b}(u)\right]=\int_0^r \frac{1}{c} f_p\left(\frac{t}{c}\right)\left(1-\frac{t}{r}\right) d t$$
> > >
> > > Section 4 discusses these properties and their theoretical guarantees extending these to our UGC framework. In the experiment section 5 we can see the empirical validation of these theorems and hence the superiority of the UGC framework.
> > >
> > >
> > > We appreciate the reviewer's reference to the Graph Summarization paper. However, it's essential to highlight that while the mentioned paper provides valuable insights, much of the discussed works lack theoretical guarantees and they are computationally expensive. Notably, these methods were proposed before the advent of GNN models, and they often do not include evaluations on downstream tasks with GNNs. In our paper, we have incorporated the most recent advancements in attributed graph coarsening [JMLR 23, JMLR 2020, ICML 23, KDD 21] to address these limitations.
> > >
> > >
> > > [JMLR 23] Kumar, Manoj, Anurag Sharma, and Sandeep Kumar. "A Unified Framework for Optimization-Based Graph Coarsening." Journal of Machine Learning Research 24.118 (2023): 1-50.
> > >
> > > [KDD 21] Huang, Zengfeng, et al. "Scaling up graph neural networks via graph coarsening." Proceedings of the 27th ACM SIGKDD conference on knowledge discovery & data mining. 2021
> > >
> > > [JMLR 20] Loukas, Andreas. "Graph Reduction with Spectral and Cut Guarantees." J. Mach. Learn. Res. 20.116 (2019): 1-42.
> > >
> > > [ICML 23] Kumar, Manoj, et al. "Featured Graph Coarsening with Similarity Guarantees." (2023).
> > >
> > > **Ans for re R5 and re R6)**
> > >
> > > We thank reviewer for the clarification.
> > >
> > > We have dedicated a whole Section 4 to define the "resemble" and "quality" of coarsened graph. The spectral and cut guarantees built in Loukas work are done using the $\epsilon-$similarity measure please see Section 3 page 10 in [1]. [1] used eigenvectors of G as the $x$ in $\epsilon-$similarity ,Leveraging on this we have established a more stringent $\epsilon \leq 1$ bound for UGC by using augmented feature vectors. The $\epsilon$-similarity is inherently linked to the eigenvalues and eigenvectors of the graph, emphasizing the significance of Relative Eigenvalue Error (REE). The probabilistic bound is encapsulated in Theorem 4.1, directly derived from Locality-Sensitive Hashing (LSH), with empirical results validating this theorem in Section 5. REE, HE, and $\epsilon$-similarity serve as quality measures and are widely adopted in various graph coarsening methods to assess the accuracy of the coarsened graph. Our experimental results in the Experiment section confirms that UGC excels across all these quality measures, establishing the superiority of the coarsened graph learned by our framework.
> > >
> > > If the reviewer sees fit, we are open to adopting the title: "Universal Graph Coarsening with Preserved Spectral Properties."
> > >
> > > [1] Loukas, Andreas. "Graph Reduction with Spectral and Cut Guarantees." J. Mach. Learn. Res. 20.116 (2019): 1-42.
> > >
> > > [2] Kumar, Manoj, Anurag Sharma, and Sandeep Kumar. "A Unified Framework for Optimization-Based Graph Coarsening." Journal of Machine Learning Research 24.118 (2023): 1-50.

---

> > > > ### Comment · Reviewer_rN8j · 2023-11-22
> > > >
> > > > I thank the authors for their continued responses and for correcting equation (1) to now reflect the actual set of "coarsening" matrices. I will take those responses into account during the reviewer discussion period and will then decide to change my score.

---

### Official Review · Reviewer_aXHZ · 2023-11-08

**Soundness:** 3 good
**Presentation:** 4 excellent
**Contribution:** 3 good
**Rating:** 6
**Confidence:** 3

**Summary:**

The authors propose a graph coarsening algorithm that works in heterophilic scenarios by borrowing from locality sensitive hashing (LSH) literature. To do so, they consider the node feature vector in addition to the adjacency matrix. The core idea is to map nodes using this augmented matrix to the same node using an appropriately instantiated LSH. To validate the quality of the coarsening, the authors consider relative eigen error, hyperbolic error, and, to account for the feature vector, bounded $\epsilon$-similarity. They show computational gains over previous coarsening state-of-the-art in terms of memory and compute time. To demonstrate the applicability of their approach, the authors train a single hidden layer GCN, and show that training on the coarsened graph has negligible impact on accuracy while benefiting from computational gains when tested on the original graph for predictions.

**Strengths:**

- The approach seems intuitive and there is a simplicity appeal to an adjacency augmented node feature vector.
- The results seem promising (even if the downstream task is more illustratory than extensive)

**Weaknesses:**

- Section 3.4 is not a time complexity analysis, and no appropriate appendix is present. Crucially, it is unclear how the hidden loop on Line 14 in Algorithm 1 is maintained as O(N) rather than O(N^2). A time complexity analysis section should not leave finding all hidden loops in mixed pseudo-code and mathematical notation algorithms as an exercise for the reader. The empirical results indeed hint that the time cost benefits exist, but as the linear time claim is made in the abstract and introduction, this section requires considerable improvement.
- The choice of downstream GNN architecture feels unjustified. Why specifically convolution instead of GIN, or GAT? More concretely, how should a reader know that the coarsening benefits are not specific to GCN but rather more universally exploitable? Additionally, low eigen error is shown as a benefit, but no investigation was made to show if, for example, the coarsening maps to community detection when performed using spectral methods.

**Questions:**

[Repeated from Weaknesses]
- Why specifically convolution instead of GIN, or GAT? More concretely, how should a reader know that the coarsening benefits are not specific to GCN but rather more universally exploitable?

- Additionally, low eigen error is shown as a benefit, but no investigation was made to show if, for example, the coarsening maps to community detection when performed using spectral methods.

---

> ### Author Response · Authors · 2023-11-17
> **Eagerly waiting for your feedback**
>
> Thank you for reviewing the paper and providing feedback! We have responded below to individual quotes from your review.
>
> **Ques 1)** Section 3.4 is not a time complexity analysis, and no appropriate appendix is present. Crucially, it is unclear how the hidden loop on Line 14 in Algorithm 1 is maintained as O(N) rather than O(N^2). A time complexity analysis section should not leave finding all hidden loops in mixed pseudo-code and mathematical notation algorithms as an exercise for the reader. The empirical results indeed hint that the time cost benefits exist, but as the linear time claim is made in the abstract and introduction, this section requires considerable improvement
>
> **Ans 1)**  We have three phases for our framework. In the first phase(Algo 1 Line 1-7), we can see Line 7 is driving the complexity of the algorithm where we multiply two $F \in \mathbb{R}^{N \times d}$ and $W \in \mathbb{R}^{L \times d}$ matrices which results to $O(NLd)$. In the second pass, the supernodes for the coarsened graphs are constructed with the help of the accumulation of nodes in the bins. The main contribution of UGC is up to these two phases i.e., Line 1-10. Till now, time-complexity is $O(NLd) \equiv O(NC)$ where $C$ is a constant. Hence, the time complexity for getting the partition matrix is $O(N)$.
>
> In the third phase, Line 14-15, we calculate the adjacency and features of the supernodes of the coarsened graph $G_{c}$. For this we iterate over the edges of the original graph and use the edge nodes along with the surjective mapping $\pi : V \rightarrow V_c$ to increment the weight of the corresponding edge between the supernodes in $G_{c}$. The computational cost of this operation is $O(m)$, where $m$ is the number of edges in the original graph, and this is a one time step. Indeed, the overall time complexity of all three phases combined is O(N+m) where m is the number of edges. However, it's important to note that the primary contribution of UGC lies in the process of finding the partition matrix whose time complexity is O(N). We have  compared the partition matrix computational of all other methods with ours.
>
> We have added this Detailed Time-complexity analysis in Appendix also.
>
> **Ques 2)** The choice of downstream GNN architecture feels unjustified. Why specifically convolution instead of GIN, or GAT? More concretely, how should a reader know that the coarsening benefits are not specific to GCN but rather more universally exploitable?
>
> **Ans 2)** We thank the reviewer for their constructive suggestions. While we initially employed GCN for validating the quality of our coarsened graph, it's essential to note that our model is not constrained to any specific architecture. Responding to the suggestion, we have tested our framework on additional models and results from three models (GraphSage, GIN, and GAT) are incorporated here. These results validate that our framework is not model dependent. If the reviewer deems it beneficial, we are open to add these results into Appendix.
>
> Models are trained on 50% coarsened graph and accuracies are quoted on original data.
> | Model\Data | Cora     | Pubmed   | Physics  | Squirrel |
> | --------   | -------- | -------- | -------- | -------- |
> | GCN        | 89.30    | 84.77    | 96.12    | 31.62    |
> | GraphSage  | 69.39    | 85.72    | 94.49    | 61.23    |
> | GIN        | 67.23    | 84.12    | 85.15    | 44.72    |
> | GAT        | 74.21    | 84.37    | 92.60    | 48.75    |
>
>
> **Ques 3)**
>
> Additionally, low Eigen error is shown as a benefit, but no investigation was made to show if, for example, the coarsening maps to community detection when performed using spectral methods.
>
> **Ans 3)** We thank the reviewer for their feedback. To elucidate the correlation between Eigen Error(REE) and the quality of the coarsened graph, we draw the reviewer's attention to Figure 5(a). In this figure, it is evident that the Relative Eigen Error (REE) increases as we reduce the graph. This observed trend is consistent with the relationship observed in node classification accuracy using GNNs, where the accuracy decreases as the graph gets reduced. These findings are corroborated in Table 6, presented in Appendix A.5. The presented results collectively illustrate the intrinsic connection between Relative Eigen Error, GNNs accuracy, and the quality of the coarsened graphs.

---

> > ### Comment · Reviewer_aXHZ · 2023-11-18
> >
> > Thank you for your response and clarification.
> >
> > Regarding Ans3, thank you for the clarifications.
> >
> > Regarding Ans1, thank you for including this in the appendix, I do think this is needed within the paper.
> >
> > Regarding, Ans2, I did suspect that the results should be model-agnostic, but I did want to see at least empirical evidence towards this as well. I think the paper would be improved if these results were included in the Appendix as well.

---

> > > ### Author Response · Authors · 2023-11-18
> > >
> > > We thank reviewer for the feedback.
> > >
> > > Regarding Ans2, We have added model-agnostic results in Appendix.
> > >
> > > **We hope the proposed changes, additional experiments and clarifications will convince the reviewer of the merits of our work. If the reviewer agrees, we humbly appeal to please consider increasing the score.**

---

> > > > ### Author Response · Authors · 2023-11-23
> > > >
> > > > Dear Reviewer aXHZ,
> > > >
> > > > Thank you for your feedback. We have carefully addressed all your comments in the rebuttal, making substantial improvements to the paper. **We would appreciate if you could reconsider your scores on the paper in the light of the responses to the comments of all reviewers, in case you feel it deserves to be in..**
> > > >
> > > > Best regards,

---

### Meta-Review · Area_Chair_4MrA · 2023-12-08

**Metareview:**

**Summary**
This paper studies the problem of simplifying graphs, aiming at reducing a potentially large input graph into a small-scale graph with fewer nodes and edges. The proposed algorithm first constructs an augmented feature matrix for a graph, incorporating node-level features, structure-level features, and heterophily, which represents the proportion of edges between nodes of different classes. Subsequently, the augmented feature matrix is transformed into a reduced graph using LSH. The efficiency and the effectiveness of the proposal have been empirically evaluated on real-world datasets.

**Strengths**
- The problem studied in this paper is well-motivated and relevant. As large-scale graphs become increasingly common, simplifying them while retaining core information is crucial for downstream tasks.
- The technique of combining feature matrix argumentation and LSH appears to be novel and interesting.

**Weaknesses**
- The presentation needs significant improvement. Even after revision by the authors, there are still numerous typos and grammatical mistakes. For example, in the very first definition of a graph, "$|N|$" should be "$N$" in "$V = \\{v_1, v_2, \dots, v_{|N|}\\}$", and $(V \times V)$ should be $V \times V$. Although the definition of $S$ in Equation (1) has been improved, it is still not perfectly correct as variable and predicate parts are not separated by some symbol like ":" or "|", and it is hard to parse it.
- There are several unclear points, as highlighted by the reviewers, such as the evaluation protocol, which deteriorates the overall quality and the significance of the paper. While some of these concerns have been partially addressed in the authors' response, I am not convinced that all of them have been fully incorporated into the submission.

**Justification For Why Not Higher Score:**

The identified weaknesses in the paper, as outlined above, are crucial and must be addressed for the publication of this paper. Through discussion with reviewers, we have reached a consensus that the clarity issue is crucial. In conjunction with other weaknesses raised by the reviewers, I have decided to reject the paper. Since the technical contribution of this paper is potentially interesting, I strongly advise addressing all the raised issues by the reviewers for substantial improvement before resubmission.

**Justification For Why Not Lower Score:**

N/A

---

### Decision · Program_Chairs · 2024-01-16

Reject